# The Spatiotemporal Expression of Notch1 and Numb and Their Functional Interaction during Cardiac Morphogenesis

**DOI:** 10.3390/cells10092192

**Published:** 2021-08-25

**Authors:** Lianjie Miao, Yangyang Lu, Anika Nusrat, Hala Y. Abdelnasser, Sayantap Datta, Bin Zhou, Robert J. Schwartz, Mingfu Wu

**Affiliations:** 1Department of Pharmacological and Pharmaceutical Sciences, College of Pharmacy, University of Houston, Houston, TX 77204, USA; lmiao4@central.uh.edu (L.M.); ylu37@central.uh.edu (Y.L.); anusrat2@CougarNet.UH.EDU (A.N.); habdelna@CougarNet.UH.EDU (H.Y.A.); sdatta20@CougarNet.UH.EDU (S.D.); 2Department of Genetics, Albert Einstein College of Medicine of Yeshiva University, New York, NY 10461, USA; bin.zhou@einsteinmed.org; 3Department of Biology and Biochemistry, University of Houston, Houston, TX 77204, USA; sjrobert@central.uh.edu

**Keywords:** notch signaling, numb family proteins, cardiac progenitor cell differentiation, outflow tract, cardiomyocyte proliferation

## Abstract

Numb family proteins (NFPs), including Numb and Numblike (Numbl), are commonly known for their role as cell fate determinants for multiple types of progenitor cells, mainly due to their function as Notch inhibitors. Previous studies have shown that myocardial NFP double knockout (MDKO) hearts display an up-regulated Notch activation and various defects in cardiac progenitor cell differentiation and cardiac morphogenesis. Whether enhanced Notch activation causes these defects in MDKO is not fully clear. To answer the question, we examined the spatiotemporal patterns of Notch1 expression, Notch activation, and Numb expression in the murine embryonic hearts using multiple approaches including RNAScope, and Numb and Notch reporter mouse lines. To further interrogate the interaction between NFPs and Notch signaling activation, we deleted both *Notch1* or *RBPJk* alleles in the MDKO. We examined and compared the phenotypes of *Notch1* knockout, NFPs double knockout, *Notch1*; *Numb*; *Numbl* and *RBPJk*; *Numb*; *Numbl* triple knockouts. Our study showed that Notch1 is expressed and activated in the myocardium at several stages, and Numb is enriched in the epicardium and did not show the asymmetric distribution in the myocardium. Cardiac-specific *Notch1* deletion causes multiple structural defects and embryonic lethality. *Notch1* or *RBPJk* deletion in MDKO did not rescue the structural defects in the MDKO but partially rescued the defects of cardiac progenitor cell differentiation, cardiomyocyte proliferation, and trabecular morphogenesis. Our study concludes that NFPs regulate progenitor cell differentiation, cardiomyocyte proliferation, and trabecular morphogenesis partially through Notch1 and play more roles than inhibiting Notch1 signaling during cardiac morphogenesis.

## 1. Introduction

Numb, an intracellular adaptor protein, was identified as the first cell fate determinant. This protein is responsible for distinguishing the cell fate of sibling cells by asymmetric distribution and by inhibiting Notch signal. Numb’s function as an inhibitor of Notch1 signaling during the development of the peripheral and central nervous system and muscle cell differentiation has been indicated from genetic evidence in *Drosophila.* [1]. Numb also regulates cardiac progenitor cell differentiation in *Drosophila* [2,3], and in Zebrafish, it controls the heart tube laterality [4].

Mammalian Numb (Nb) and its homolog Numblike (Nl), collectively known as Numb family proteins (NFPs), being expressed ubiquitously during embryogenesis [5], function in determining neural stem cell fate as well as regulating its differentiation [6,7]. NFPs are involved in the specification and differentiation of hematopoietic stem cells [8], muscle satellite cells [9], cancer stem cells [10], and hemangioblasts [11]. They function in a conserved manner within the mammalian Notch1 pathway [5,6,12,13]. For example, the overexpression of mammalian Numb antagonizes Notch1-dependent transactivation of the *Hes1* promoter [14] and impedes Notch1 activity in neurite growth [15]. Like its counterpart in *Drosophila*, mammalian Numb is asymmetrically distributed in dividing precursor cells and is preferentially segregated to one daughter cell upon cell division, thus securing their individual cell fate through the suppression of Notch signaling [12,16,17,18,19,20,21,22].

Multiple studies have reported that NFPs regulate many biological processes of cardiac morphogenesis, including epicardial development, outflow tract (OFT) alignment/septation, atrioventricular septation, cardiac progenitor cell differentiation, cardiomyocyte proliferation, myocardial trabeculation, and ventricular compaction [23,24,25,26,27,28,29,30]. The epicardium, the outer layer of the heart, is composed of a single layer of epicardial cells. Conditional deletion of NFPs specifically in the epicardium causes the disruption of the epicardial adherens junction and epicardial polarity and randomizes spindle orientations, leading to epicardial cell epithelial–mesenchymal transition (EMT) defects and embryonic lethality [8]. 

NFPs also regulate the morphogenesis of OFT. OFT is formed by several developmentally distinct cell populations, including cardiomyocytes and endothelial cells derived from the second heart field (SHF) and vascular smooth muscle cells (SMC) [31]. SHF gives rise to the right ventricle, OFT, interventricular septum, endocardium, and part of the inflow region [32,33,34,35,36,37]. Perturbation of SHF progenitor cell deployment and differentiation leads to a spectrum of congenital heart diseases (CHDs). SHF is distinguished from the first heart field by expressing *Isl1* [38], *Tbx1* [39], *Fgf8*, and *Fgf10* [33,37,38,40,41]. Myocardial NFPs double knockout (MDKO) mediated via *Nkx2.5^Cre/+^* displayed defects in OFT alignment, OFT septation, and atrioventricular septation. NFPs double deletion mediated by *Mef2c-Cre*, which is active in SHF [36], recapitulated the morphogenetic defects in MDKO. *aMHC-Cre*, which is active in cardiomyocytes at a later stage than *Nkx2.5^Cre/+^*, mediated NFP deletion did not cause defects in OFT morphogenesis [23,26]. These reports indicate that the morphogenesis defects in MDKO are due to cardiac progenitor cell differentiation defects. 

Several signaling molecules and transcriptional factors including Fgf [33,41,42], Wnt [43,44,45,46], Hedgehog [47,48], Tbx1 [49], Notch [50], BMP [51], and retinoic acid [52] are involved in the deployment of SHF progenitor cells to the elongating linear heart tube and subsequent Isl1 cell differentiation [53,54,55]. NFPs double deletion mediated by *Nkx2.5^Cre/+^* resulted in higher expression of SHF progenitor markers, such as *Isl1*, *Tbx1*, *Fgf8*, and *Shox2*, and the knockouts displayed abnormal expression levels of cardiomyocyte maturation/differentiation markers, such as *Myh6*, *Myh7*, *Bmp10*, and *Irx3-5*. One *Notch1* allele deletion in MDKO did not fully rescue the differentiation and OFT morphogenetic defects in MDKO [23]. Whether both *Notch1* alleles or *RBPJk* alleles are required for NFPs to regulate SHF cardiac progenitor cell differentiation and cardiac morphogenesis is unknown. 

In this study, to determine the genetic and functional interactions between NFPs and Notch, considering that NFPs’ inhibiting Notch1 occurs in the cell that expresses both Numb and Notch1, we firstly examined the spatiotemporal patterns of Notch1 expression, Notch activation, and Numb expression in the embryonic hearts during cardiac morphogenesis using RNAScope, a Notch reporter [56], and a Numb reporter [28]. Contrary to previous reports, we found that Notch1 is weakly expressed and activated in the myocardium at several stages. Cardiac-specific *Notch1* deletion causes multiple structural defects and embryonic lethality. We did not observe an asymmetric distribution of Numb in the myocardium but did observe its apparent accumulation in the epicardium. To further interrogate the interaction between NFPs and Notch signaling activation during cardiac morphogenesis, we deleted both *Notch1* [57] or *RBPJk* [58] alleles in the MDKO. We examined and compared the phenotypes of *Notch1* single knockout, *RBPJk* single knockout, NFPs double knockout, *Notch1*; *Numb*; *Numbl* triple knockouts (TKO), and *RBPJk*; *Numb*; *Numbl* triple knockouts (TKO-R). We found that *Notch1* or *RBPJk* deletion in the MDKO did not rescue the structural defects of OFT in the MDKO, and surprisingly, the TKO died at younger ages than the MDKO. We found that the defects of cardiac progenitor cell differentiation, trabecular morphogenesis, and the expression of cell cycle regulator p57 were partially rescued in TKO. However, the fact that the defects, especially structural defects, are not fully rescued in TKO or TKO-R, suggests that NFPs play other roles beyond inhibiting Notch1 signaling during cardiac morphogenesis, which is supported by the differential expression profiles between MDKO and TKO examined by mRNA deep sequencing. Our study concludes that NFPs regulate progenitor cell differentiation, cardiomyocyte proliferation, and trabecular morphogenesis partially through Notch1 and regulate the interaction between cells and extracellular matrix in addition to inhibiting Notch1 signaling during cardiac morphogenesis.

## 2. Materials and Methods

### 2.1. Mouse

Mouse strains *Numb^fl/fl^*, *Numbl^fl/fl^* [59,60], transgenic Notch reporter H2B-Venus [56], *RBPJk* [58], and *Notch1^fl/fl^* [57] were obtained from Jackson Lab. Dr. Robert Schwartz provided *Nkx2.5^Cre/+^* [61] mice. *Nkx2.5^Cre/+^*; *Numb^fl/+^*; *Numbl^fl/fl^* or *Nkx2.5^Cre/+^*; *Numb^fl/fl^; Numbl^fl/+^* males were mated to *Numb^fl/fl^*; *Numbl^fl/fl^* females to generate *Nkx2.5^Cre/+^*; *Numb^fl/fl^*; *Numbl^fl/fl^* designated as MDKO; Nkx2.5^Cre/+^; Numb^fl/+^; *Numbl^fl/fl^*; *Nt1^fl/+^* or Nkx2.5^Cre/+^; Numb^fl/+^; *Numbl^fl/fl^*; *RBPJk^fl/+^* males were mated to *Numb^fl/fl^*; *Numbl^fl/fl^*; *Nt1^fl/fl^* females to generate *Nkx2.5^Cre/+^*; *Numb^fl/fl^*; *Numbl^fl/fl^*; *Nt1^fl/+^* designated as MDKO; *Nt1^fl/+^* and *Nkx2.5^Cre/+^*; *Numb^fl/fl^*; *Numbl^fl/fl^*; *Nt1^fl/fl^* designated as TKO or Nkx2.5^Cre/+^; Numb^fl/+^; *Numbl^fl/fl^*; *RBPJk^fl/fl^* designated as TKO-R, their sibling embryos without the *Cre* allele are designated as controls. All animal experiments were approved by the Institutional Animal Care and Use Committee (IACUC) at the University of Houston and performed according to the NIH Guide for the Care and Use of Laboratory Animals. The protocol number is PROTO202000060 and the latest approval date is 9 September 2020. 

### 2.2. Paraffin and Frozen Section Immunofluorescence

Immunofluorescence (IF) and Hematoxylin and Eosin (H&E) staining were performed as described [23]. Briefly, embryos or heart samples were fixed in 4% PFA for 2 h at room temprature or overnight at 4 °C. After fixation, samples were washed with PBS and embedded in OCT. Then the sample blocks were sectioned at 10 μm per section. After sectioning, slides were immersed in PBS for 10 min to get rid of OCT. The sections then were permeabilized with PBT (0.5% Tween in PBS) (if needed) and then blocked for 30 min with TNB blocking buffer (Perkin Elmer, FP1020, Waltham, MA, USA) at RT. After blocking, the sections were incubated with primary antibodies diluted in blocking buffer overnight at 4 °C. Then, the slides were washed with PBT 3 × 10 min at RT and followed by secondary antibodies incubation at RT for 1 h. After secondary antibody incubation, the sections were counterstained and mounted in mounting medium (Vectashield, H-1000-10, Burlingame, CA, USA) for confocal imaging. The following primary antibodies were used: Endomucin (1:100 Santa Cruz, sc-65495, Dallas, TX, USA), MF20 (1:100, Developmental Studies Hybridoma Bank (DSHB), MF20, Iowa, IA, USA), BrdU (1:50; Becton Dickinson, 347583, Franklin Lakes, NJ, USA), PECAM (1:50; BD Pharmingen, 550274, San Diego, CA, USA), N1ICD (1:50; Cell Signaling, 4147S, Danvers, MA, USA), Numb (1:400, Santa Cruz, H-70; or 1:600, Cell Signaling, 2756s), P57 (1:200, Abcam, ab75974, Cambridge, UK). 

### 2.3. Cardiomyocyte Proliferation Assay via BrdU Pulse Labeling

Pregnant females were intraperitoneally injected with BrdU for 1 h before harvesting the embryos. The proliferation rate was assessed by the percentage of BrdU positive cardiomyocytes out of total cardiomyocytes following our previous protocol. Cardiac troponin T and Endomucin were stained to distinguish cardiomyocytes from non-cardiomyocytes.

### 2.4. mRNA In Situ Hybridization

Single mRNA molecule in situ hybridization (ISH) and Immuno-fluorescent staining (IFS) were performed according to the protocol of the kit RNAscope 2.5 HD (RED) Assay (Advanced Cell Diagnostics, Cat. No. 322360, Newark, CA, USA) and our published protocol [62], which enables the detection of single mRNA molecules. Briefly, after fixation for 24 h, the embryos were frozen embedded in OCT compound. Sections of the frozen embedded samples were processed following the protocol of the kit. The mRNA expression level in each cell was determined based on the number of mRNA molecules or signal intensity using the confocal scanned pictures, and three scanned sections for each cell were quantified [62].

### 2.5. Imaging

The following systems were used. For confocal imaging, Zeiss LSM 880-NLO confocal microscope system (Jena, Germany) with an Airyscan detector with a FAST module on a Zeiss Axio observer Z1 inverted microscope equipped with an internal spectral QUASAR detector. Stereo images of the heart or embryos were harvested by a stereoscope (Leica M205 FA, Wetzlar, Germany).

### 2.6. Western Blot Analysis

Western blot was performed as previously described [23]. Briefly, E12.5 hearts were harvested and lysed in RIPA buffer. Protein concentration was determined using the BCA kit (Thermo Fisher, 23225, Waltham, MA, USA) and equal amounts were run on SDS-PAGE using 4–20% Mini-PROTEAN^®^ TGX™ Precast Protein Gels (Bio-Rad, 4561093, Hercules, CA, USA) and transferred onto PVDF membranes (GE Healthcare Life Science, 10600023, Chicago, IL, USA) following standard protocols. The antibodies used in the study include Numb (1:1000, Cell Signaling, 2756S), GAPDH (1:1000, Santa Cruz Biotechnology, sc25778), N1ICD (1:1000; Cell Signaling, 4147S), and P57 (1:1000, Abcam, ab75974).

### 2.7. Trabecular Density and Thickness Quantification and IF Staining

One in every four sections from E12.5 hearts were stained with Endomucin and MF20 antibodies. The number of trabeculae per unit length and the thickness of trabeculae or compact zone in both left and right ventricles were determined through the quantification of at least six sections from the medial part of the heart. Trabecular thickness was stated as the average width of the base, middle and top of an individual trabecula. Compact zone thickness was defined as the average width of six different spots in left and right ventricle in each section. The trabecular length and thickness were measured using the software *LAS X Life Science Microscope Software* from Leica.

### 2.8. RNA Isolation and Quantitative PCR (Q-PCR)

For Q-PCR analysis, Aurum Total RNA Mini Kit (Bio-Rad, 732-6820) and the RNeasy Micro Kit (Qiagen, Hilden, Germany) were used to isolate the total amount of RNA from the three whole hearts for each experiment. The experiments were repeated at least thrice for each age. Reverse transcription (iScript cDNA Synthesis Kit, Bio-Rad) was carried out with at least 300 ng of total RNA. The gene expressional level was analyzed using standard Q-PCR methods with iTAQ SYBR Green Master Mix on a CFX96 instrument (Bio-Rad). Each sample was run in triplicate and normalized to cyclophilin A. Primer sequences utilized are listed in Appendix A.

### 2.9. Statistical Analysis

All values are presented as mean ± s.e.m. Differences between groups were compared by Student’s *t*-test. *p* < 0.05 was considered significant.

## 3. Results

### 3.1. Spatiotemporal Expression and Activation of Notch1 during Cardiac Morphogenesis

Previous studies have revealed two mechanisms of mammalian Numb inhibiting Notch signaling. NFPs promote the ubiquitination of the Notch1 receptor and the degradation of the Notch1 intracellular domain (N1ICD), circumventing its nuclear translocation and downstream activation of Notch1 target genes [14]. It was also reported that Numb regulates post-endocytic trafficking and degradation of Notch1 [63]. These studies suggest that Numb–Notch interaction occurs in the same cell. Therefore, it is necessary to examine the spatiotemporal expression patterns of Notch1 and Numb and the activation of Notch during cardiac morphogenesis. *Notch1* global knockout hearts display more severe cardiac morphogenetic defects than the knockout hearts of other Notch receptors [64,65]. Therefore, we mainly examined the expression of Notch1 by RNAScope, which can detect mRNA molecules at a single molecule resolution [66], and its activation by staining Notch1 intracellular domain (N1ICD), the active form of Notch1 [67]. At E9.5, we found that Notch1 is mainly expressed in the endocardial cells, consistent with the previous study [68]. Interestingly, *Notch1* is also weakly expressed in the ventricular cardiomyocytes (Figure 1A,A1) and cardiac progenitor cells in OFT (Figure 1B,B1). Notch1 is not detected in ventricular cardiomyocytes at E10.5 and E11.5 (Figure 1C,C1 and data not shown) but in cells of OFT at E10.5 (Figure 1D,D1). Starting at E12.5, ventricular cardiomyocytes resume to express *Notch1*, and its expression at E13.5 increased to two times higher than that at E12.5 (Figure 1E,F). This is consistent with a previous report that N1ICD is detected in cardiomyocytes in E13.5 and older hearts [69]. In addition to its strong expression in the endocardial cells at E9.5 and E10.5 (Figure 1G,H), N1ICD is detected in the cardiac progenitor cells in OFT at E9.5 and E10.5 (Figure 1H and data not shown) but is not detectable in the cardiomyocytes at E10.5 (Figure 1G). 

The single-cell-resolution Notch reporter line CBF: H2B-Venus [56] was applied to monitor Notch activation. In addition to strong activation in the endocardium (Figure 1I,J), we found that Notch is weakly activated in the cardiac progenitor cells in OFT and some ventricular cardiomyocytes at E8.5 and E9.5 (Figure 1I,J and data not shown), a similar observation in zebrafish [70].

### 3.2. Spatiotemporal Expression of Numb during Myocardial Morphogenesis

Considering that NFPs inhibit Notch signaling in a cell-autonomous manner, we then examined the transcriptional pattern of Numb using RNAScope and then determined its expression pattern via a Numb reporter mouse line [28]. We found that Numb is expressed in epicardial cells, endocardial cells, and cardiomyocytes at E10.5 (Figure 2A). The specificity of *Numb* in situ hybridizations via RNAScope is confirmed by the *Nkx2.5^Cre/+^* mediated *Numb* deletion in the myocardium (Figure 2B). We further examined its expression at E12.5 and found its transcription in the endocardial cells and epicardial cells to be relatively higher as compared to the cardiomyocytes in the myocardium (Figure 2C). The relative enrichment in endocardial cells can be observed in the MDKO, as the expression of *Numb* in the myocardium is deleted (Figure 2D). To determine the expression pattern of Numb, a mCherry:Numb knockin line was applied. In this line, mCherry was inserted right after the start codon of Numb so that its expression can be detected by mCherry [28]. We found that Numb is not enriched in pro-epicardial cells at E9.5 and epicardial cells at E10.5, but is enriched in epicardial cells at E12.5 and later stage (Figure 2E–G). The enrichment of Numb in epicardial cells at E12.5 and later stage but not in the pro-epicardial cells is consistent with the previous report that NFPs are required for epicardial cell EMT, but not epicardial development [30]. Numb is ubiquitously expressed in trabecular and compact cardiomyocytes and did not display an asymmetric distribution between them (Figure 2G). 

Numb is well known for promoting progenitor cell differentiation by its asymmetric distribution in dividing cells, e.g., Numb is asymmetrically distributed in the mitotic epicardial cells [30]. Previous studies show that NFPs regulate cardiac progenitor differentiation [23,25] and a potential mechanism is its asymmetric distribution during cardiac progenitor cell division. Therefore, we examined Numb distribution in the cells of OFT, in which cardiac Isl1 progenitor cells localize and where Isl1 progenitor cells will differentiate to Isl1 negative and MF20 positive cardiomyocytes (Figure 2F). We found that Numb did not display asymmetric distribution between MF20 positive and MF20 negative cells in the OFT (Figure 2F), although it is highly expressed in the endoderm (Figure 2F). We also examined Numb in dividing and non-dividing progenitor cells in OFT and cardiomyocytes in ventricle at E9.5 and did not find its asymmetric distribution in these cells (data not shown), suggesting that Numb does not regulate cardiac progenitor cell differentiation through its asymmetric distribution during cardiac morphogenesis.

### 3.3. Notch1 Is Required for Outflow Tract Clockwise Rotation and Septation

To further study the genetic and functional interactions between NFPs and Notch, we generated *Notch1* knockout (N1KO), *RBPJk* knockout (RKO), NFPs double knockout (MDKO), *Notch1*; *Numb*; *Numbl* triple knockout (TKO), and *RBPJk*; *Numb*; *Numbl* triple knockout (TKO-R). A previous study shows that *cTnt-Cre* mediated deletion of Notch1 in cardiomyocytes did not cause any structural and functional defects in the mice [71]. In this study, our data indicate that Notch1 is expressed and activated in cardiac progenitor cells at E9.5 and ventricular cardiomyocytes at E9.5, E12.5, and older (Figure 1). *Mef2cCre* is also active in the cardiac progenitor cells of the SHF, but not in the cardiomyocytes of left ventricle, which prevents the study of trabecular morphogenesis in the left ventricle. Therefore, we deleted *Notch1*, *Numb*, and *Numbl* via *Nkx2.5^Cre/+^*, which is active in cardiac progenitor cells and all the cardiomyocytes [61]. We compared the defects of the knockouts and found that N1KO died before E17.5 (Table 1). The hearts are relatively smaller with a shorter distance between the base and the apex and display defects in OFT morphogenesis with an abnormal alignment between the pulmonary artery and aorta at E13.5 (Figure 3A). The ventricles also display a less deep left-right ventricular groove (Figure 3A,D,E), suggesting their role in maintaining standard ventricular structure. We examined the cardiac structural defects in more detail through H&E staining. The N1KO hearts did not display defects in trabecular morphogenesis but in OFT morphogenesis (Figure 3B,C). The OFT shows a rotation defect at the base of the aorta and pulmonary artery, although the distal portion presents normal septation (Figure 3A–C). Furthermore, the atrioventricular valvular formation was affected with a thick AV cushion (Figure 3D,E). The N1KO hearts display no obvious defects in the trabecular formation (Figure 3D,E). We examined the N1KO at E15.5 and found that N1KO hearts display defects in OFT alignment (Figure 3F,G) and no apparent flaws in the ventricular compaction but show abnormalities in AV valvular morphogenesis (Figure 3H,I) (N = 5). Previous studies have shown that NFPs are required for the OFT morphogenesis. Whether NFPs and Notch1 regulate OFT morphogenesis in a coordinated way will be examined in later sessions. 

### 3.4. Notch1 or RBPJk Deletion in MDKO Did Not Rescue the Structural Defects of MDKO 

Previous studies have shown that MDKO hearts display multiple morphogenetic defects and die at around E14.5 [23]. To determine whether Notch signaling activation causes the defects in MDKO and whether Notch suppression can rescue the defects of MDKO, we deleted both *Notch1* alleles or *RBPJk* alleles in MDKO. We first examined the survival rate and found that the percentages of live TKO are smaller than the expected percentages starting at E9.5, and we did not harvest any TKO beyond E14.5 (Table 2). The survival rates indicate that TKO died earlier than MDKO, suggesting that *Notch1*, *Numb*, and *Numbl* work synergistically to regulate the survival. We examined the defects of OFT alignment and septation and AV valvular morphogenesis in TKO. The TKO, like MDKO, display defects of OFT alignment and septation (Figure 4A–D) and atrioventricular septation defect (AVSD) (Figure 4E–H) examined at E12.5 (n = 5). Since other Notch receptors are also expressed in the myocardium, we determined if the deletion of *RBPJk*, a transcriptional co-repressor required to bind to NICD and activate canonical Notch downstream targets [72], in MDKO can rescue the defects. Surprisingly, TKO-R hearts, like the MDKO hearts, display defects in atrioventricular septation and OFT alignment and septation (Figure 4I–L) (N = 5). 

### 3.5. NFPs Regulate Progenitor Cell Differentiation and Trabecular Morphogenesis Partially through Notch1 

One of the significant defects in MDKO is the progenitor differentiation, as the MDKO hearts display significantly higher expression of *Isl1*, *Fgf8*, and *Fgf10*, markers of SHF progenitor cells [23]. A previous publication shows that Notch1 is involved in the inhibition of cardiac progenitor differentiation [73]. To determine whether NFPs regulate the cardiac progenitor cell differentiation through *Notch1*, we examined the expression of *Isl1*, *Fgf8*, and the OFT structural defects in MDKO, MDKO; *Nt1^+/−^*, and TKO. We found that *Isl1* expression is significantly reduced in MDKO; *Nt1^+/−^*, and TKO at E10.5 (Figure 5A). The *Isl1* expression in MDKO; *Nt1^+/−^* was reduced to not significantly different from control, and its expression in TKO was significantly lower than the control (Figure 5A). Notch signaling regulates the level of *Fgf8* [74] and *Fgf8* is significantly up-regulated in MDKO (Figure 5B) [23]. We therefore examine if Notch regulates *Fgf8* via *Notch1*. The level of *Fgf8* is significantly reduced in MDKO, *Nt1^+/−^* and TKO hearts compared to control and MDKO (Figure 5B). Surprisingly, the OFT alignment defect and septation defects in MDKO, *Nt1^+/−^*, and TKO are not rescued in all the hearts we examined (Figure 4). These results suggest that a rescue of *Isl1* and *Fgf8* expression is insufficient to rescue the OFT structural defects, and NFPs might regulate multiple other factors during OFT morphogenesis. To determine potential factors that NFPs might interact with, we performed mRNA deep sequencing to compare the differential gene expression profiles between MDKO and TKO hearts at E11.5 (Appendix A). We then performed gene-ontology analysis and found that genes/components that are involved in extracellular matrix (ECM), extracellular region/space, collagen trimer, fibrinogen complex, and collagen–ECM interaction are either down or up-regulated (Appendix A), suggesting that NFPs play roles in remodeling the organization of ECMs and their interactions with the cells. We also analyzed the biological processes involving the down- or upregulated genes in MDKO compared to TKO. We found that biological processes, such as ECM organization, response to stimulus or chemicals, and cellular organization are enriched (Appendix A). 

We also examined the defects of trabecular morphogenesis in the hearts of the four different genotypes (Figure 5C–F). The trabecular and compact zones were identified by staining with Endomucin/PECAM, which stains endocardial and endothelial cells, and MF20, which stains cardiomyocytes. The trabecular density (identified by the number of trabeculae per unit length), relative trabecular thickness, and the relative width of the compact zone were measured and compared among the hearts of the four genotypes. We found that trabecular density and trabecular thickness are partially rescued when one or both Notch1 alleles were deleted in MDKO (Figure 5G,H). The thickness of the compact zone is not significantly different among the four types of hearts at E12.5 (Figure 5I). 

### 3.6. NFPs Regulate p57 Partially through Notch1

CDKN1c (*p57^kip2^* or *p57*) is repressed by Notch1 [75] and *RBPJk* deletion promotes *p57* expression [68]. The activated Notch1 signaling in endocardial cells promotes cardiomyocyte proliferation by inhibiting *p57* through Bmp10 in the myocardium [68,76,77,78]. NFPs deletion significantly reduces *p57* expression and increases cardiomyocyte proliferation [23], prompting us to hypothesize that NFPs regulate *p57* by repressing Notch1 signaling. Thus, we tested whether deletion of both *Notch1* alleles in MDKO would abolish the *p57* down-regulation and reduce the proliferation rate in the MDKO. *p57* expression was examined and compared at both transcriptional and translational levels between MDKO and TKO hearts. Consistently, *p57* expression is reduced in MDKO at E10.5 based on Q-PCR, one or both *Notch1* alleles deletion in MDKO increases the *p57* expression (Figure 6A) but *p57* expression level does not increase to the level of control (Figure 6A). We also examined the P57 by immunostaining and found that P57 was expressed mainly in endocardial cells at E12.5 (Figure 6B). The expression of P57 in MDKO and TKO was reduced as the percentage of P57 positive endocardial cells was reduced to ~40% (Figure 6B–E). We then examined the P57 expression via Western blot using embryonic hearts from the same litter. The P57 expression in MDKO; *Nt1^+/−^* and TKO was reduced based on Western blot (Figure 6F,G; Appendix A). We also applied BrdU pulse labeling to examine and compare the cardiomyocyte proliferation rate in trabecular and compact zones among hearts with different genotypes. Trabecular cardiomyocytes in both MDKO; *Nt1^+/−^* and TKO display a significantly lower proliferation rate compared to MDKO, but a significantly higher rate than the control at E12.5 (Figure 6H–N), suggesting that NFPs regulate cardiomyocyte proliferation partially through Notch1 and NFPs also regulate cardiomyocyte proliferation in a Notch1 independent manner.

## 4. Discussion

### 4.1. NFPs Play Multiple Roles in a Cell Type-Dependent Manner during Cardiac Morphogenesis 

NFPs’ functions during cardiovascular morphogenesis have been revealed recently. Multiple studies show that NFPs regulate many biological processes in a cell type-dependent manner, i.e., NFPs regulate epicardial development, endothelial cell-mediated angiogenesis, cardiac progenitor cell differentiation, cardiomyocyte proliferation, outflow tract alignment/septation, atrioventricular septation, myocardial trabeculation, and ventricular compaction [23,24,25,26,27,28,29,30]. 

The function of NFPs in endocardial cells was examined by deleting NFPs in endocardial cells via the Tie2-Cre. The knockout did not display apparent defects [23], suggesting that the roles of NFPs in shaping the heart structure are independent of their functions in endocardial cells. Whether NFPs’ deletion in Tie2 expressing cells will affect angiogenesis is not extensively examined. Indeed, a study shows that NFPs regulate VEGF receptor endocytosis, signaling, and recycling in endothelial cells to promote the angiogenic growth of blood vessels [79]. The epicardium, as the outer layer of the heart, is forged from a single layer of epicardial cells. Conditional deletion of NFPs specifically in the epicardium results in the disruption of the epicardial adherens junction and epicardial polarity and randomizes spindle orientations, leading to epicardial cell EMT defects and embryonic lethality [8]. NFPs regulate the morphogenesis of OFT, as MDKO hearts displayed defects in OFT alignment/septation and atrioventricular septation. NFPs double deletion mediated by Mef2c-Cre, which is active in SHF, recapitulated the morphogenetic defects in MDKO [23]. aMCH-Cre, which is active in cardiomyocytes at a later stage compared to Nkx2.5^Cre/+^, and SM22Cre mediated NFP deletion in cardiomyocytes did not cause defects in OFT morphogenesis [23,26]. These reports indicate that the OFT morphogenic defects in MDKO might be due to differentiation defects of the cardiac progenitor cells. This is supported by the abnormal expression of the markers of SHF cardiac progenitor cells in MDKO hearts. Indeed, another study shows that NFPs regulate differentiation and self-renewal of the cardiac progenitor cells in the SHF [25]. The current study shows that *Notch1* allele deletion in MDKO partially rescues the expression of *Isl1* and *Fgf8*, suggesting that NFPs regulate cardiac progenitor cell differentiation via Notch1. Consistently, previous studies show that Notch1 in cardiac progenitor cells and cardiomyocytes in the OFT plays an essential function in the dilation of the aortic root, and this function is independent of its role in endothelial cells [80,81]. 

NFPs also regulate trabecular morphogenesis and ventricular compaction. The MDKO hearts display defects in trabecular formation with a smaller number of trabeculae per unit length and thicker trabecula at early embryonic stage [23]. Further cellular mechanistic study shows that NFPs null cardiomyocytes display a loss of cellular orientation during trabecular initiation due to a loss of membrane-localized N-Cadherin [28], a molecule that regulates cellular behaviors during trabeculation in zebrafish [82]. The tracing of single NFPs null cell shows that NFPs regulate the cardiomyocyte migration and proliferation in a cell-autonomous manner during trabecular formation [28]. NFPs also regulate left ventricular compaction at a later stage, as the MDKO hearts display prominent trabeculae and thick compact zone [23,26,27]. A potential mechanism of NFPs regulating myocardium compaction is through inhibiting Notch2. Notch2 intracellular domain (N2ICD) is detected throughout the myocardium before E11.5. In contrast, at a later stage, Notch2 activity is limited to the trabecular zone in the myocardium during ventricular compaction, and it is specifically down-regulated in the compact zone areas.

Yang et al. [26] indicated that Notch2 might be involved in myocardium compaction. Notch2 global knockout hearts display ventricular hypoplasia [65] and its deletion via SM22-Cre causes cyanosis at birth due to a narrow artery, and whether or not the knockout hearts display trabeculation defect has not been reported [83]. NFPs deletion might abolish the asymmetric activation pattern of Notch2 and cause noncompaction and hyper-trabeculation defect in MDKO [26]. Ventricular noncompaction and hyper-trabeculation have been increasingly recognized clinically, which are determined as congenital myocardial defects that might be associated with disrupted development of the ventricular wall [84]. Genetic analysis of patients, mostly with isolated forms of ventricular noncompaction without CHDs, has suggested a broad genetic heterogeneity, with a complex and multifactorial network contributing to the etiology and pathogenesis [84]. Currently, surviving patients carrying NFPs mutations have not been reported, although point mutations of Numb were recorded in several databases, including http://www.ncbi.nlm.nih.gov/clinvar/ (accessed on 20 August 2020). Given the pivotal role of NFPs in ventricular wall formation and the embryonic lethal phenotype of NFPs-deficient mice, human patients with germline loss-of-function mutations for both Numb and Numbl would seem to have little chance of survival in utero. Therefore, the redundant function of Numb and Numbl reduces the probability of finding LVNC patients that carry NFPs mutations. 

### 4.2. NFPs Regulate Cardiac Morphogenesis via Multiple Molecules/Signaling Pathways

During cell division, Numb, as an intracellular protein being segregated asymmetrically, influences cell fate [17] and inhibits Notch signaling [7,16,20,85]. Numb inhibits Notch to regulate progenitor cell differentiation in *Drosophila*, while in vertebrates, the relationship between Numb and Notch1 during embryogenesis remains unclear and controversial [7,12]. The phenotype of the NFPs global knockout is similar to that of Notch1 pathway disrupted mutants, which leads to the hypothesis that Numb and Notch1 signaling are connected. However, Notch1 targets were not up-regulated in the NFP global knockout during mammalian development, as would be predicted if Numb inhibits Notch1 signaling [86]. Indeed, previous studies regarding Notch activation are not consistent in MDKO hearts [24,26,27,30]. The exact Numb–Notch1 relationship during cardiac morphogenesis needs further investigations. 

Notch1 is activated when the signal-receiving cells are adjacent to the signal-sending cells. When both Notch1 and Numb are expressed in the same cell(s), then there is a possibility that Numb could inhibit Notch1 signaling. Our studies show that Notch1 is not expressed in the cardiomyocytes at all stages (Figure 1), as Notch1 at E10.5 and E11.5 in cardiomyocytes could not be detected by RNAScope. This might explain why Numb mediated inhibition of Notch was not detected in one of the studies [27]. Furthermore, genetic tools such as the epistatic status between NFPs and Notch1 should be applied to reveal their genetic and functional interactions. In this study, we deleted both alleles of Notch1 in MDKO. *Isl1* and *Fgf8* were significantly up-regulated, while *p57* was significantly down-regulated in MDKO. With one *Notch1* allele deletion, *p57* transcriptional level was increased. However, with both *Notch1* alleles’ deletion, *p57* transcriptional level was not increased to the level of wildtype, suggesting that *p57* was also regulated by other molecules/signaling pathways. *Isl1* and *Fgf8* were significantly down-regulated in TKO compared to MDKO. Our data clearly demonstrate that NFPs regulate the expression of multiple genes, including *Isl1*, *p57*, and *Fgf8*, via Notch signaling. Genetic epistasis showed that Notch1 up-regulation is partially responsible for the lower *p57* expression, higher cardiomyocyte proliferation rate, and the increased thickness of trabeculae. However, the fact that the deletion of both alleles of *Notch1* or *RBPJk* in MDKO did not fully rescue many of the defects of MDKO suggests that NFPs regulate cardiac morphogenesis through other signaling pathways as well. 

This is consistent with many other reports that NFPs regulate multiple biological processes via different molecules and signaling pathways. Numb is a cargo-selective endocytic adaptor that performs clathrin-dependent endocytosis through mediating the cargoes attachment to the clathrin adaptor α-adaptin [87]. It appears that one of the major mechanisms for NFPs regulating these biological processes is endocytosis and trafficking [28,63,88,89]. NFPs regulate N-Cadherin membrane localization via endocytosis to regulate cellular orientation and, subsequently, trabecular initiation and growth [28]. NFPs regulate the activities of Erbb2 and Stat5 via affecting the endocytosis progression during trabecular morphogenesis [27]. NFPs regulate the intracellular destination and stability of the Notch ligand Delta-like 4 (Dll4) through a post-endocytic-sorting process [90]. Numb directly interacted with p120 catenin (p120), followed by association with E-cadherin, and prevented its internalization in regulating tissue morphogenesis and cell polarity.

According to Sato et al. [91], Numb regulates the sorting of Notch1 through late endosomes for degradation, and depletion of Numb facilitates Notch1 recycling [63]. Many other proteins that are regulated by NFPs are not listed here, and NFPs might regulate the endocytosis and degradation of many unknown proteins too. Therefore, the phenotypes that we observed in NFPs knockout might be the consequence of many distorted pathways due to the absence of NFPs. 

In the future, it will be interesting to identify the potential proteins that interact with Numb or are regulated by Numb through mass-spectrometry. Furthermore, it is essential to identify the endocytic functions of Numb by deleting the three NPF (Asn-Pro-Phe) or DPF (Asp-Pro-Phe) motifs in the Numb in vivo. Transcriptional profiles of MDKO and TKO suggest that NFPs regulate biological processes, such as ECM organization, response to stimulus or chemicals, and cellular organization, and determining how NFPs regulate these processes will be essential to understand their functions in cardiac morphogenesis.

## Figures and Tables

**Figure 1 cells-10-02192-f001:**
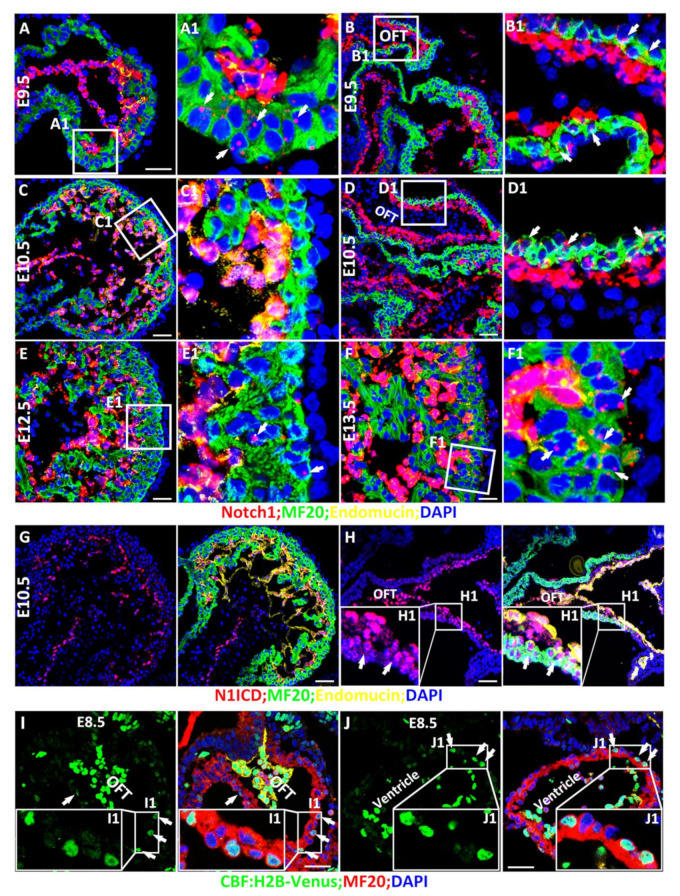
Spatiotemporal expression and activation of Notch1 during cardiac morphogenesis. (**A**,**B**) Notch1 was mainly expressed in the endocardial cells, and weakly expressed in the ventricular cardiomyocytes indicated by arrows; (**A**,**A1**) and cardiac progenitor cells in OFT indicated by arrows; (**B**,**B1**). (**C**,**D**) Notch1 is not detected in ventricular cardiomyocytes (**C**,**C1**) but cells in OFT indicated by arrows) at E10.5 (**D**,**D1**). (**E**,**F**) Starting at E12.5, ventricular cardiomyocytes resume expressing Notch1 (indicated by arrows; (**E**,**E1**), and its expression increased significantly at E13.5 indicated by arrows (**F**,**F1**). (**G**,**H**) Notch1 activation was detected by N1ICD IF staining and consistently, N1ICD is strongly expressed in the endocardial cells. N1ICD is also detected in some of the cardiac progenitor cells in OFT indicated by arrows with weak signaling (**H**,**H1**) but not in the cardiomyocytes at E10.5 (**G**). (**I**,**I1**,**J**,**J1**) A Notch reporter line CBF: H2B-Venus Notch1 showed that Notch is not only activated in the endocardium but also weakly activated in the cardiac progenitor cells in OFT (**I**,**I1**) and some ventricular cardiomyocytes (**J**,**J1**) at E8.5. Arrows point to the cells that expression Notch1 or N1ICD. Scale bars: 50 μm.

**Figure 2 cells-10-02192-f002:**
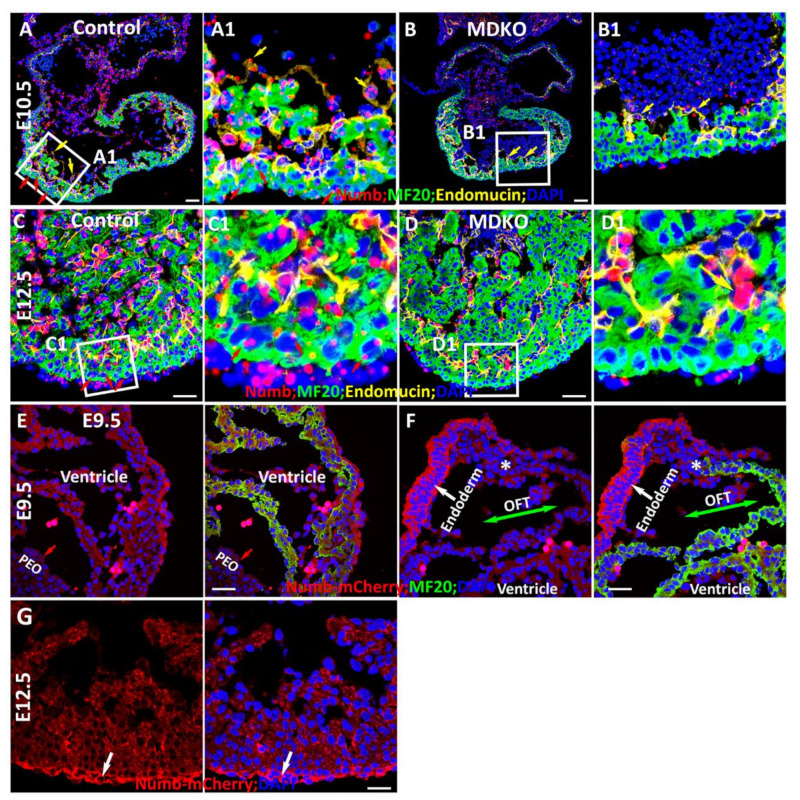
Spatiotemporal expression of Numb during myocardial morphogenesis. (**A**(**A1**)–**D**(**D1**)) Numb is expressed in epicardial cells, endocardial cells and cardiomyocytes at E10.5 and E12.5 (**A**(**A1**),**C**(**C1**)). The specificity of Numb in RNAScope and relative enrichment in endocardial cells is confirmed and observed in the MDKO (**B**(**B1**),**D**(**D1**)). Red arrows point to the cardiomyocytes and yellow arrows point to the endocardial cells in (**A**–**D**). (**E**,**F**) The Numb expression pattern was further determined by a mCherry::Numb knockin line. Numb is not enriched in pro-epicardial cells at E9.5 (**E**) but is enriched in epicardial cells at E12.5 (**F**). Numb is ubiquitously expressed in trabecular and compact cardiomyocytes and did not display an asymmetric distribution between them (**E**,**F**). Epicardial cells are enriched with Numb (**G**). Red or white arrows point to the cells that express Numb. * Indicates a transition from MF20 negative cells to MF20 positive cells in OFT. Scale bars: 100 μm.

**Figure 3 cells-10-02192-f003:**
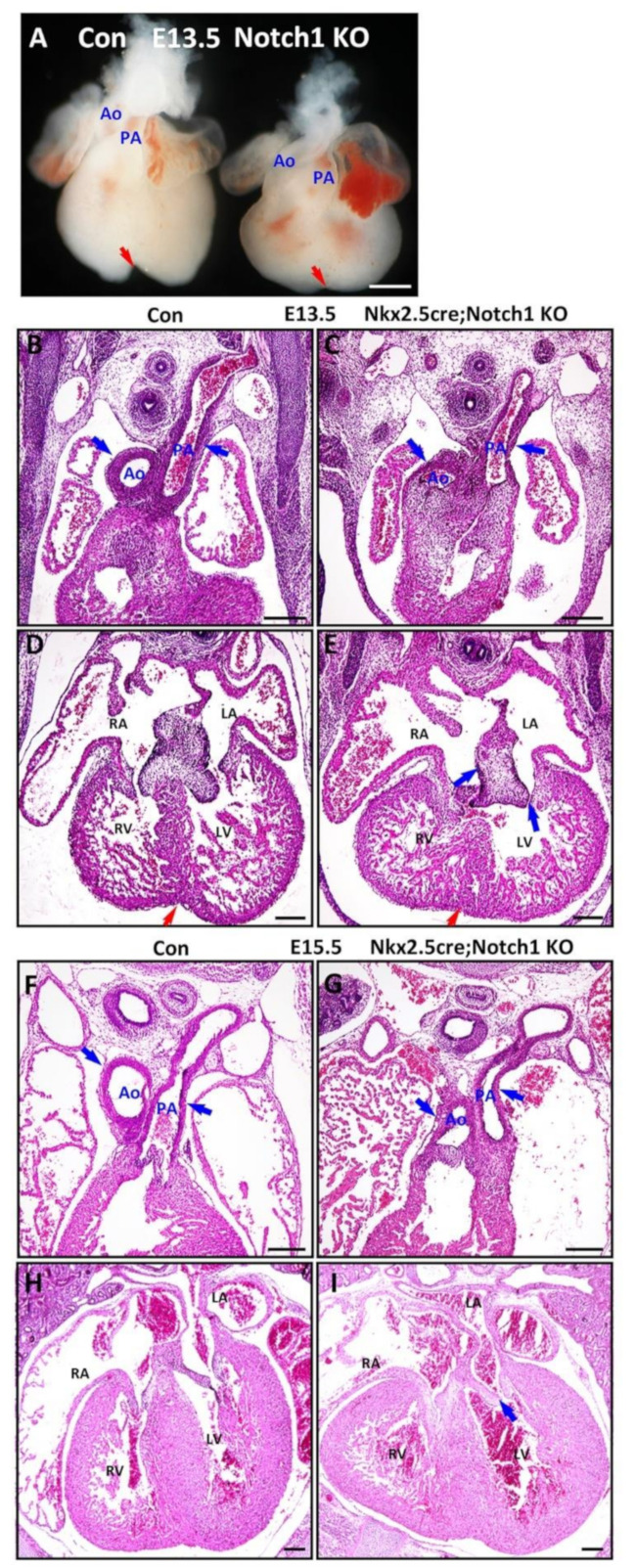
Notch1 is required for outflow tract clockwise rotation and septation. (**A**–**E**) The myocardial Notch1 knockout hearts via Nkx2.5^Cre/+^ are relatively smaller and display abnormal alignment between pulmonary artery and aorta at E13.5 (blue arrows), and the knockout hearts also display a less deep left-right ventricular groove (red arrows). (**F**–**I**) At E15.5, the N1KO hearts displayed OFT alignment defects but no obvious ventricular compaction defects. The blue arrows indicate aorta (Ao), pulmonary artery (PA) and their valves, red arrows indicate ventricular groove. Scale bars: 100 μm.

**Figure 4 cells-10-02192-f004:**
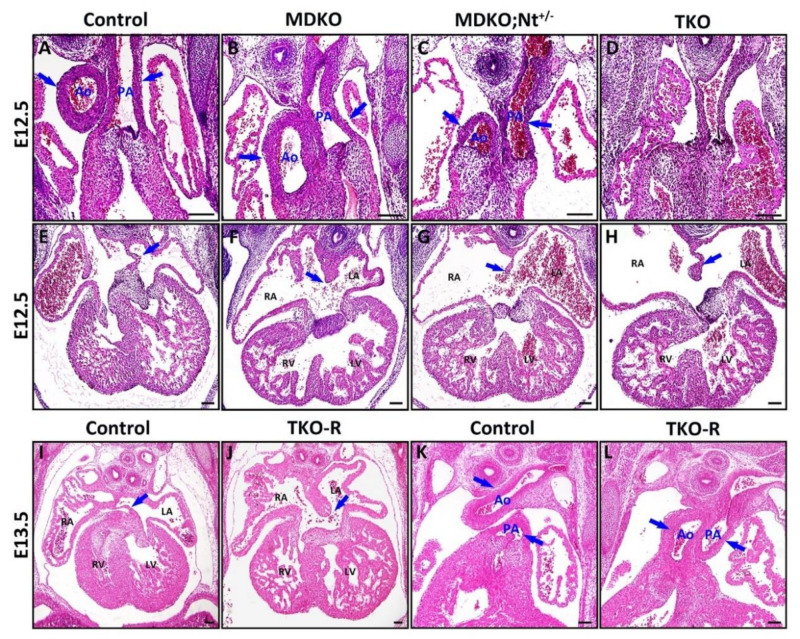
Notch1 or RBPJk deletion in MDKO did not rescue the structural defects of MDKO. (**A**–**H**) The morphologies of Control, MDKO, MDKO; Nt1^+/−^, and TKO were examined via H&E staining at E12.5. Both MDKO and TKO display OFT alignment and septation defects of (**A**–**D**) and atrioventricular septation defect (AVSD) (**E**–**H**) (n = 5). (**I**–**L**) *Numb*, *Numbl* and *RBPJk* triple knockout (TKO-R) hearts display AVSD and OFT alignment and septation defect (N = 5). The blue arrows indicate aorta (Ao), pulmonary artery (PA) or atrial septum. Scale bars: 200 μm.

**Figure 5 cells-10-02192-f005:**
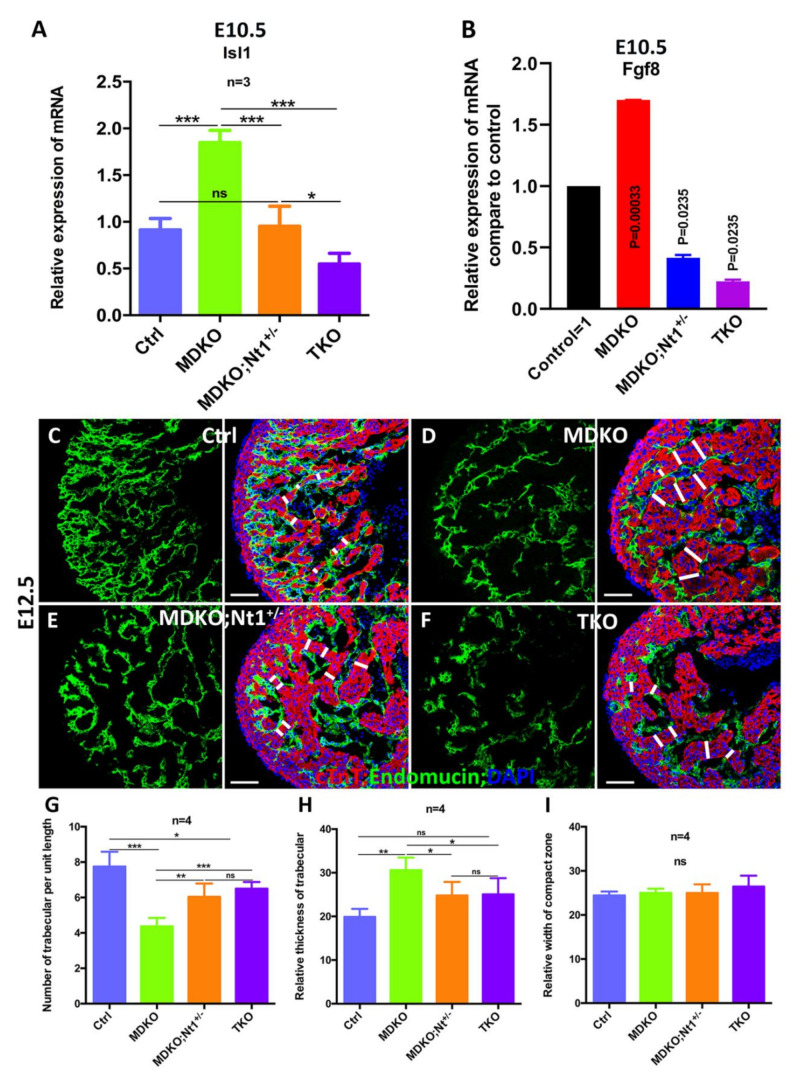
NFPs regulate progenitor cell differentiation and trabecular morphogenesis partially through Notch1. (**A**) *Isl1* expression is significantly reduced in TKO compared to MDKO at E10.5 based on Q-PCR. (**B**) *Fgf8* expression is significantly reduced in TKO compared to MDKO at E10.5 based on Q-PCR. (**C**–**I**) show that MDKO hearts display trabeculation defects with thicker trabeculae and a smaller number of trabeculae per unit length (**D**,**G**–**I**). One or both alleles deletion of Notch1 partially rescued the trabeculation defects that appeared in MDKO (**E**–**I**). ***** 0.01 < *p* < 0.05, ** 0.001 < *p* < 0.01, *** *p* < 0.001. ns, not significant. Scale bars: 100 μm.

**Figure 6 cells-10-02192-f006:**
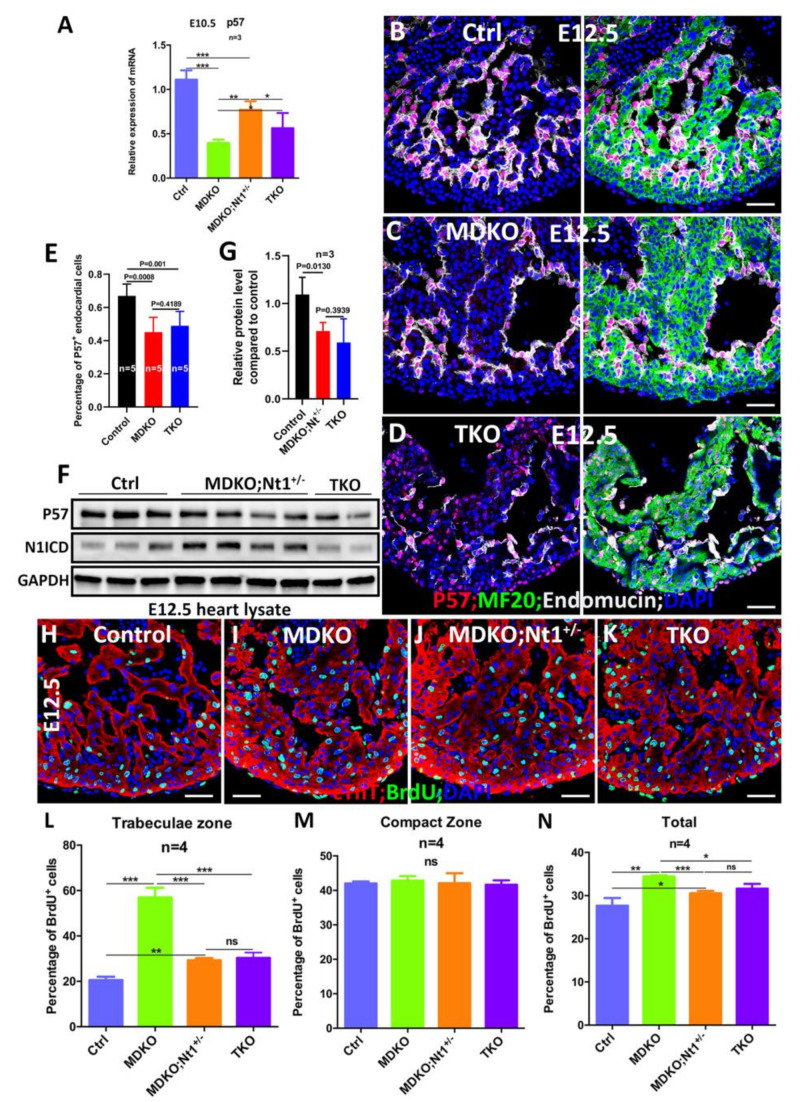
NFPs regulate p57 partially through Notch1. (**A**) P57 is downregulated in MDKO, and the abnormal expression of p57 was partially corrected in TKO based on Q-PCR. (**B**) P57 was mainly expressed in endocardial cells at E12.5. (**C**–**E**) The expression of p57 in MDKO and TKO was reduced compared with control. (**F**–**G**) The p57 expression was reduced in both MDKO; *Nt1^+/−^* and TKO based on western blot. (**H**–**N**) BrdU pulse labeling showed that cardiomyocyte proliferation increased in MDKO trabecular zone (**I**,**L**) compared with control (**H**,**L**). Cardiomyocytes proliferation rate in MDKO; *Nt1^+/−^* and TKO trabecular zone decreased significantly compared to that in MDKO, but is still higher than that in control (**J**–**L**). Compact zone cardiomyocytes proliferation rate among four types of hearts didn’t show a significant difference (**H**–**K**,**M**). * 0.01 < *p* < 0.05, ** 0.001 < *p* < 0.01, *** *p* < 0.001. ns, not significant. Scale bars: 100 μm.

**Table 1 cells-10-02192-t001:** The survival rate of *Nkx2.5^cre/+^*; *Notch1^fl/fl^* (KO).

Age	Total Embryos	KO	Harvested/Expected Percentage of KO
E9.5	75	17	90.67
E10.5	52	24	184.62
E11.5	7	2	114.29
E12.5	35	6	68.57
E13.5	59	15	101.69
E14.5	26	5	76.92
E15.5	26	5	76.92
E16.5	41	6	58.54
E17.5	2	0	0.00
Postnatal	42	0	0.00

**Table 2 cells-10-02192-t002:** Notch1 deletion in MDKO causes earlier death of the embryos.

Age	Total Embryos	MDKO;Nt1^+/−^	TKO	Harvested/Expected Percentage of MDKO;Nt1^+/−^	Harvested/Expected Percentage of TKO	Harvested/Expected Percentage of MDKO
E9.5	204	25	18	98.04	70.59	100.00
E10.5	439	53	32	96.58	58.31	100.00
E11.5	149	15	11	80.54	59.06	100.00
E12.5	577	59	31	81.80	42.98	76.00
E13.5	173	17	8	78.61	36.99	68.00
E14.5	44	3	0	54.54	0.00	40.00

## Data Availability

All the data are presented in this manuscript. Unpublished data is available upon request. The mouse lines that we generated are available upon request.

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
