# Peer review of "The Spatiotemporal Expression of Notch1 and Numb and Their Functional Interaction during Cardiac Morphogenesis"

_cells, 2021, doi:10.3390/cells10092192_

Round 1

Reviewer 1 Report

Recently, several studies showed that Numb family proteins (NFPs) modulate diverse biological processes, i.e., epicardial development, endothelial cell-mediated angiogenesis, progenitor cell differentiation, OFT septation etc. This manuscript is well thought and nicely written. Although, it looks little bit descriptive in result and discussion section. Overall, it has the merit for publication in this journal. Authors suggested that Notch1 is activated when the signal-receiving cells are adjacent to the signal sending cells and there is a possibility that Numb could inhibit Notch1 signaling in an autocrine or paracrine fashion. They also found that Notch1 is not expressed in the cardiomyocytes at all stages of embryonic development. At what age Notch1 may be expressed in cardiomyocytes?

Authors demonstrated that Isl1 and Fgf8 were up-regulated, where as p57 is significantly down-regulated in MDKO where both alleles of Notch1 were deleted. Although, with one Notch1 allele deletion, p57 transcriptional level was increased. How authors can explain this phenomenon. They suggested involvement of some other molecules/signaling pathways. What could be the signaling molecules?

The immunohistochemistry and immunofluorescence protocols are very crucial, and several times do not generate reproducible results. Authors should provide details protocol for these experiments. “Reference-90” does not show method details.

Similarly, details protocol for western blotting or any reference is missing.

How was the trabecular density and thickness measured? Authors may provide software details/references.

This current manuscript did not provide the qPCR primer list/supplementary files. Authors may consider the Bio-Rad verified primers which are commercially available for their qPCR experiments.

In Fig.6F, the western blots were reprobed for N1ICD/GAPDH or separate gels were run for each antibody with same amount of loading proteins? Authors may provide full blots with molecular weight markers for each blot.  

Author Response

Please find our response to Reviewer #1's comment  in the file uploaded!

Reviewer 2 Report

This study evaluated a number of combinations of Numb and Notch family mutations. The expression images are beautiful, the phenotypic analyses were performed rigorously, and the authors should be applauded for what is clearly a very substantial amount of mouse work. The primary insight is the clarification that NFPs have functions in heart development larger than simply related to negative regulation of Notch signaling. Although the nature of these alternative processes is not identified in the current study, this is worthwhile information and the genetic approach taken provides a solid foundation from which later studies might then clarify these functions. I have some relatively minor concerns with the presentation or interpretation of certain aspects of the study, which are listed below mostly in order of appearance, but overall I thought this was a solid study.

  1. Since Fig 1 A-F uses the same markers, I suggest authors put the label for all underneath, just as done for G-H and I-J.

  1. Line 205, Fig 1B is described as showing Notch1 expression in cardiac progenitor cells in the OFT. I don’t think that these are progenitor cells at this point, they are already expressing differentiated markers (including MF20 as shown here) and are irreversibly committed to (if they haven’t already reached) their terminal fate. To properly evaluate SHF progenitors, sections through the splanchnic mesoderm (where the SHF is) would be needed. A similar statement about progenitor cells is made related to N1ICD on line 212, I cannot be certain from the image where this is in the OFT but it probably also is not in the splanchnic mesoderm. Same with comment on line 227, and elsewhere through the manuscript.

  1. Section 3.1, Notch1 is newly described as being expressed in cardiomyocytes, based on Fig 1A-F. In all cases, the labeling pattern is punctate, and seemingly always over the nucleus. This gives the appearance of being nuclear transcription that perhaps is not exported to the cytoplasm. The pattern of N1ICD shown in Fig 1H indicates that there indeed is detectable protein expression, so this is not of high concern, but should be presented clearly to avoid confusion.

  1. Line 212, authors refer to cells in the OFT and indicate Fig 1G, but the OFT is in Fig 1H.

  1. Fig 1I-J, authors could consider a higher magnification view of the Venus+ cardiomyocytes, these appear to be only on the outside of the myocardium and could possibly be another cell type, a higher magnification view would probably resolve this.

  1. Line 235-237, I am puzzled by this comment. First, doesn’t Nkx2.5-Cre cause recombination in endocardium and myocardium, and perhaps also in epicardium? I think there are published papers that show this. Second, there still appear to be a few Numb+ cardiomyocytes in panel B1.

  1. I have a few questions about section 1.3. First, I believe that the Nkx2.5-Cre line is a knock-in, so these mice are heterozygous for Nkx2.5, and possibly this influences phenotype. Second, the comparison is made between the absence and presence of phenotype in cTnt-Cre and Nkx2.5-Cre Notch1 mutants. If Nkx2.5 recombines in other lineages in addition to cardiomyocytes (see comment above), then this difference could be explained on the basis of gene function in a different cell type. It is only in the discussion that it is presented that endocardium and epicardium knockout have already been done, it would be appropriate to make some reference to this information in section 1.3 so that the reader is not wondering about this possibility.

  1. The text associated with the qPCR analysis of Fig 5 should make clear that this was conducted on ventricular tissue, and the methods section should further describe how this tissue was isolated in order to allow the reader to know if valve or atrial tissue was also present. The analysis is somewhat misguided, if the argument is that NFPs regulate progenitor differentiation, then why would ventricular tissue be examined here?

  1. Fig 6, endomucin is misspelled.

  1. Line 432, here in the discussion, along with mention of past use of Tie2-Cre and epicardium-Cre conditional mutation of NFPs, we now see also that Mef2c-Cre was previously studied, along with aMHC-Cre. This work was done by the current authors. So, it seems like one conclusion of the current study, that NFPs function in the SHF, was already well established. The earlier text could be revised to make clear that the present study duplicates this finding now by using Nkx2.5-Cre, and explain what advantages or insights are provided by this line compared to Mef2c-Cre.

11.  Line 538, this last sentence does not make sense, perhaps the authors meant NFP instead of NPF, DPF (or DFP) was never defined, and what are “motifs in the Numb”?

Author Response

Please find our response to Reviewer #2's comments in the uploaded file. 
